# Polar and phase domain walls with conducting interfacial states in a Weyl semimetal MoTe$_2$

Fei-Ting Huang[1], Seong Joon Lim[1], Sobhit Singh[2], Jinwoong Kim[2], Lunyong Zhang ⓘ [3], Jae-Wook Kim[1], Ming-Wen Chu[4], Karin M. Rabe[2], David Vanderbilt ⓘ [2] & Sang-Wook Cheong[1]

Much of the dramatic growth in research on topological materials has focused on topologically protected surface states. While the domain walls of topological materials such as Weyl semimetals with broken inversion or time-reversal symmetry can provide a hunting ground for exploring topological interfacial states, such investigations have received little attention to date. Here, utilizing in-situ cryogenic transmission electron microscopy combined with first-principles calculations, we discover intriguing domain-wall structures in MoTe$_2$, both between polar variants of the low-temperature($T$) Weyl phase, and between this and the high-$T$ higher-order topological phase. We demonstrate how polar domain walls can be manipulated with electron beams and show that phase domain walls tend to form superlattice-like structures along the $c$ axis. Scanning tunneling microscopy indicates a possible signature of a conducting hinge state at phase domain walls. Our results open avenues for investigating topological interfacial states and unveiling multifunctional aspects of domain walls in topological materials.

[1] Rutgers Center for Emergent Materials and Department of Physics and Astronomy, Rutgers University, Piscataway, NJ 08854, USA. [2] Department of Physics and Astronomy, Rutgers University, Piscataway, NJ 08854, USA. [3] Laboratory for Pohang Emergent Materials and Max Plank POSTECH Center for Complex Phase Materials, Pohang University of Science and Technology, Pohang 37673, Republic of Korea. [4] Center for Condensed Matter Sciences and Center of Atomic Initiative for New Materials, National Taiwan University, 106 Taipei, Taiwan. Correspondence and requests for materials should be addressed to S.-W.C. (email: sangc@physics.rutgers.edu)

In the past decade, an explosion of research has focused on a sweeping search of candidate materials that may harbor topologically protected surface states[1–6]. The appearance of massless quasiparticles near topologically protected surface states are their key features, which could be two-dimensional (2D) Dirac points on the surfaces of topological insulators (TIs), or Fermi-arc surface states attached to the bulk Weyl points in the case of three-dimensional topological Weyl semimetals (WSMs)[3–5]. The manipulation of these surface states through homo/hetero-structures between topological phases promises functionalities going beyond those of their constituents with important applications such as dissipationless electronics[7–10]. For example, when these topological insulators are interfaced with superconductors, the emergent zero-energy Majorana fermions at the boundaries can be utilized for topological quantum computation[7]. The Veselago lens, which is the electronic lenses going beyond the diffraction limit, could also be realized through Weyl semimetal $p$-$n$ junctions[9]. Despite the concept of topological protection, to utilize topological surface states remains challenging due to the chemical/structural/electronic complexity of the surfaces[11,12]. Alternatively, domain walls (DWs) of topological materials are self-assembled vacuum-free interfaces which can, in principle, replicate or facilitate new topological interfacial/edge states, but limited work has been done to date[13–15].

Among those topological materials, WSMs can be generated quite systematically in semimetallic crystals with the large spin-orbit coupling by breaking either time-reversal or space-inversion symmetry[3,5]. A considerable number of WSMs with broken inversion symmetry have been theoretically and experimentally identified, including transition-metal dichalcogenide (TMD) orthorhombic (Mo,W)(Te,P)$_2$[16–18], transition-metal monopnictide (Ta, Nb)(As, P) family[4,6,19], and the RAlGe (R = rare earth) family[20,21]. An appealing aspect of these WSMs is that they also crystallize in polar crystallographic structures with a unique polar axis along which the two opposite directions are distinguishable, and thus they are polar WSMs. Note that since they are highly conducting at low frequencies, these polar WSMs belong to the so-called polar metals that have recently drawn much attention in the ferroelectric community[22–25]. In principle, itinerant electron screening in a (semi)metal might rule out the necessity of electrostatically driven domain formation due to the fundamental incompatibility of polarity and metallicity, but the existence of polar domains, formed by local bonding preferences, is still possible since this mechanism is insensitive to the presence of charge carriers[22]. Some progress has been made in, for example, the polar interlocked ferroelastic domains in polar metal Ca$_3$Ru$_2$O$_7$[25,26] and the structural defect-mediated polar domains in metallic GeTe[27]. In this context, exploring the domain structures in polar Weyl semimetal would be particularly important because the Weyl points and Fermi-arc connectivity can be manipulated via domain reorientation or locally modified order parameters at these DWs[28–31]. Solving the Weyl equation under the experimentally known DW geometry is highly desired.

Here we choose TMD MoTe$_2$, which has recently drawn immense attention due to its phase tunability and unique physical properties, such as extremely large magnetoresistance[32], superconductivity[33,34], higher-order topology[35–37], the novel type-II WSM phase[16,17], and the polar metal (Supplementary Note 1 and Supplementary Fig. 1)[31,38,39]. Utilizing in-situ cryogenic transmission electron microscopy (TEM) and low-$T$ scanning tunneling microscope (STM), we unveil, for the first time, experimentally intriguing structures of polar domains and phase DWs between topologically distinct phases with conducting interfacial states in such a polar Weyl semimetal. We also demonstrate the real-space ferroelectric reversible switching process controlled by the electron beam of TEM. The underlying physical mechanism is understood by combing first-principles calculations and group-theoretical analysis.

## Results

**Unique layered structures of MoTe$_2$.** Depending on the crystal structure, MoTe$_2$ can be either in the semiconducting 2H or the semimetallic 1T′ phase at room temperature. 1T′-MoTe$_2$ undergoes a first-order type structural transition from a monoclinic ($P2_1/m$, space group #11) to an orthorhombic polar T$_d$ ($Pmn2_1$, space group #31) structure at a critical temperature ($T_c$) of ~260 K (Supplementary Fig. 2). T$_d$-MoTe$_2$ is a rare simultaneous example of a material with superconductivity[33,34], a polar nature, and a topologically nontrivial band structure[16,17], whereas 1T′–MoTe$_2$ is a non-polar higher-order topological material in which the 1D hinge instead of 2D surfaces host topologically protected conducting modes[35–37]. The type-II WSM transition occurs in the polar T$_d$ phase due to the requirement of broken inversion symmetry in this nonmagnetic system[16,40,41]. Notably, the inversion symmetry together with the time-reversal symmetry protects the higher-order topological phase in the nonpolar 1T′-MoTe$_2$[35–37]. Despite the apparent dissimilarity in the electronic structure, the 1T′– and T$_d$-MoTe$_2$ phases can, in fact, be considered different stacks of the similar Te–Mo–Te layers. Figure 1a illustrates the basic unit, where the off-centered Mo atoms (blue spheres) move towards each other to form metallic zigzag chains (bold red lines) running along the $a$ axis. Consequently, the Te octahedra are deformed with two possible orientations, denoted as P (**P**lus, orange arc counter-clockwise (CCW) arrow) or M (**M**inus, purple arc clockwise (CW) arrow), as shown in Fig. 1a.

Since P and M layers are translationally nonequivalent, MoTe$_2$ is a stack of P and M layers coupled through weak van der Waals forces with two possible interlayer shear displacements. Figure 1b presents the schematic of these shear displacements defined by the closest Te–Te ions (orange and purple dashed lines), denoted as + (positive; CCW displacement of Te–Te dashed line) or − (negative; CW displacement). 1T′ and T$_d$ phases can be described by the stacking sequence counting from the bottom: two 1T′ monoclinic twins are repeating arrangements of +M + P+ and −M−P− (1T′-I and 1T′-II in Fig. 1c–d), and two T$_d$-MoTe$_2$ polar states refer to stacks either as +M−P+ and −M+P− (T$_d$↑ and T$_d$↓ in Fig. 1d). Note that symbols P/M represent the intralayer displacements of Te octahedra, which remain fixed, while +/− represent the variable interlayer shifts. A T$_d$ unit consists of +/− (i.e., different) displacements of the two sides of each layer (either P or M), while a 1T′ unit has an identical interlayer displacement (either +/+ or −/− in Fig. 1c) and the preserved inversion center as marked. An uncompensated dipole, resulting in polarization along the $c$ axis, can exist in the T$_d$↑ and T$_d$↓ states due to the asymmetric Te bonding environments triggered by the interlayer shifts (Supplementary Note 1 and Supplementary Fig. 1). This subtle difference of 1T′ and T$_d$ phases has never been explicitly discussed and explains in part the effects of external strain, pressure, and thickness on phases and electronic properties of MoTe$_2$[32,33,42].

**Cross-sectional view of abundant phase domain walls.** Intriguingly, we find that 1T′ twin walls have the T$_d$ character at room temperature. A cross-section view of 1T′-MoTe$_2$ has been imaged using dark-field transmission electron microscopy (DF-TEM) in combination with high-angle annular dark-field (HAADF) scanning transmission electron microscopy (STEM) imaging, which displays strong contrast associated with the atomic number of the local composition. Figure 2a–b reveal quasi-periodic monoclinic twin domains of alternating bright and dark bands along the $c$ axis, which are consistent with the superposition of

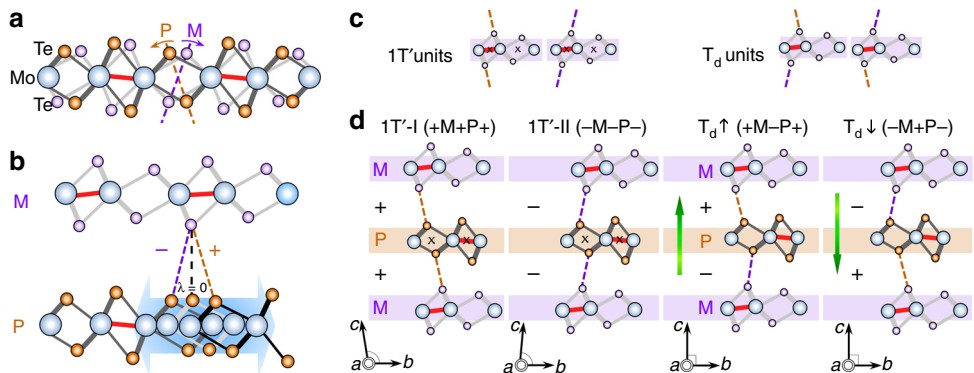

**Fig. 1** Flexible layer-structured MoTe$_2$. **a** Schematic models of a single layer of MoTe$_2$ with either P (Plus, CCW rotation) or M (Minus, CW)-type Te octahedral deformations in the cross-section view. Mo, blue; Te of a P layer, orange; Te of a M layer, purple. The red lines indicate Mo–Mo zigzags along the *a* axis. Orange and purple arc arrows represent the directions of the Te octahedral deformations. **b** Schematic models of bilayer MoTe$_2$ with P−M and P +M configurations, counting from the bottom P layer. Gliding of the bottom P layer results in + (positive)/− (negative) interlayer shifts, where the signs refer to the CCW/CW displacement of Te−Te bonding lines. A zero-interlayer shear ($\lambda = 0$) corresponds to a centrosymmetric orthorhombic reference structure T$_0$. **c** Examples of 1T′ and T$_d$ units of a M layer. A 1T′ unit requires the same sign of interlayer shearing (++ or −−) while those of a T$_d$ unit are different. **d** Three layers can glide individually to give four configurations. (1) 1T′-I, +M+ P+ with *b–c* angle of ~93.5° and (2) 1T′-II, −M−P− with *b–c* angle of ~86.5°,[56] (3) T$_d$↑, +M−P+ and (4) T$_d$↓, −M+P− with orthogonal unit cells. The polarization along the ±*c* axis (green arrows) denotes as T$_d$↑ and T$_d$↓. Note that lattice *a* and *b* of the 1T′ structure are switched to match the zigzag direction as in the T$_d$ phase (*b* > *a*). The symbol *x* marks the inversion center

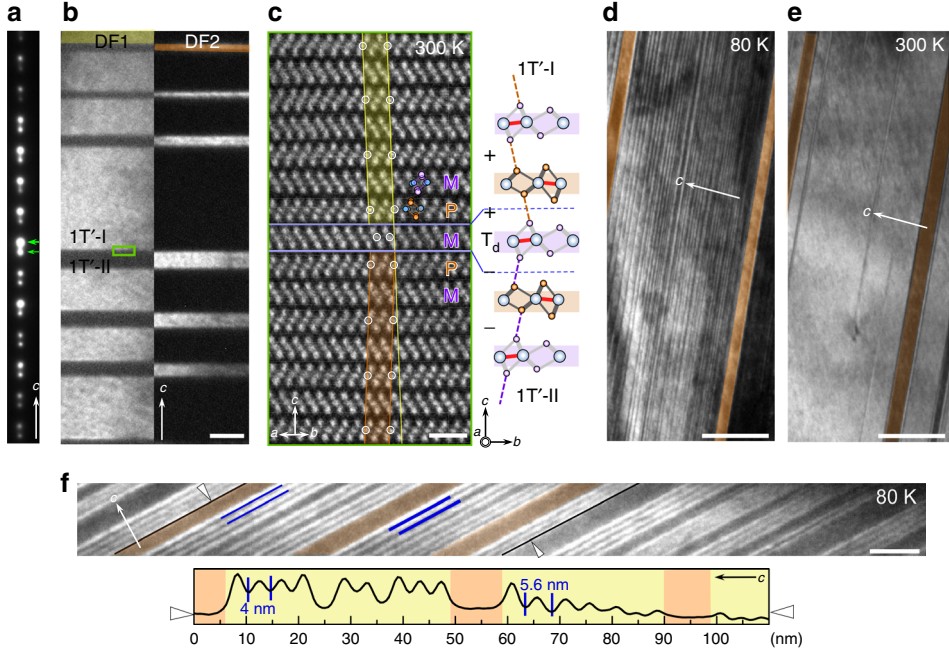

**Fig. 2** Phase domain walls along cross-section views at room and cryogenic temperatures. **a** A selected area electron diffraction (SAED) pattern of 1T′ monoclinic twins, revealing spot splitting along the *c* axis. **b** DF-images were taken using the strong and weak ($1\bar{1}2$) spots (green arrows) of variants 1T′-I and 1T′-II, denoted as DF1 and DF2, respectively. Yellow and orange false colors are added to aid the eye. The average twin width is on the order of a half-μm, and the two types of twin domains are typically unequal in size. Scale bar: 200 nm. **c** The atomic-resolved HAADF-STEM image over one twin wall along the [110] zone axis. Overlaid color-coded 1T′ unit cells defined by white-circled Mo atomic columns show a glide-reflection twin. The twin is composed of one M layer (marked by blue dashed lines) with anti-phase (+/−) interlayer shearing, connecting 1T′-I (yellow-shaded) and 1T′-II (orange-shaded) regions. The lattice model viewed along [100] is shown for clarity. Scale bar: 1 nm. **d**, **e** DF-images taken at (**d**) 80 K and (**e**) 300 K. At 80 K, the phase-separated state is observed. The appearance of thin T$_d$ layers within the initial 1T′ twins, revealing additional ($\bar{1}12$) spots due to an orthogonal T$_d$ unit cell as shown in Fig. 3a. Scale bar: 200 nm. **f** A higher-magnification DF-image, showing superlattice-like −(T$_d$)$_m$(1T′)$_n$− (*m*, *n* = integer) nanoscale phase DWs at 80 K and the corresponding intensity profile between white arrow heads along the *c* direction, covering both 1T′-I (yellow) and 1T′-II (orange) twins. The image is rotated and enlarged to enhance the display. Scale bar: 30 nm

the diffraction spots (Fig. 2a) resulting from adjacent twin domains. 1T′ twinning occurs by a mirror operation along the *ab*-plane. A further zoomed-in HAADF-STEM image of a twin wall (Fig. 2c) shows an atomically coherent interface between 1T′-I and 1T′-II along [110]. The interfaces (the blue lines in Fig. 2c)

can be readily identified by tracking the white-circled Mo positions. The yellow and orange shaded areas outline the 1T′ monoclinic unit cells above and below the interfaces. It turns out that in addition to a mirror operation, a gliding of atomic layers is imposed on either side of the twin wall to reduce lattice strain. As

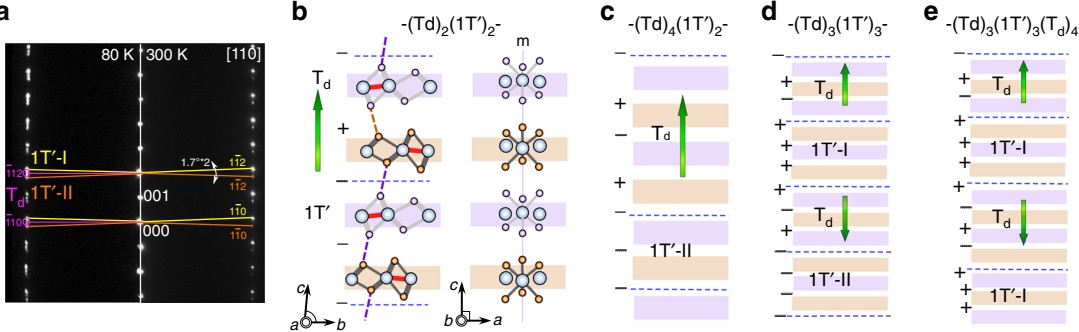

**Fig. 3** SAED pattern of MoTe$_2$ and the schematics of $-(T_d)_m(1T')_n-1T'/T_d$ superlattice along the $c$ axis. **a** The presence of 1T'-I, 1T'-II, twins and the $T_d$ phase are indicated with yellow, orange and pink straight lines. Though 1T' and $T_d$ show the same extinction rules along [110], the orthogonal and non-orthogonal angles between ($\bar{1}$10) and ($\bar{1}$12) planes indicated by pink, yellow and orange lines unambiguously can be the fingerprint of $T_d$, 1T'-I, and 1T'-II domains. **b** The smallest $-(T_d)_2(1T')_2-$ periodicity by symmetry. The configuration of $+M-P+M+P+$ corresponds to 4 layers, namely ~2.8 nm. Purple and orange blocks represent M and P layers, respectively. Blue dashed lines mark the phase DWs. The mirror symmetry, perpendicular to the $a$ axis, can be seen along [010] projection (unique axis $a$, when $b > a$). Experimentally, the thinnest periodicity is found to be 6 layers. **c** Six layers consist of either $-(T_d)_4(1T')_2-$, $-(T_d)_2(1T')_4-$ or $-(T_d)_3(1T')_3-$. Considering the $T_d$ phase as the ground state at 80 K and the lattice mismatch as explained below, these considerations lead to the most likely domain assignment $-(T_d)_4(1T')_2-$. Polar $T_d\uparrow$ can nucleate from either 1T'-I or 1T'-II without preference. The same rule is applied to $T_d\downarrow$. **d** Schematic model of the $-(T_d)_3(1T')_3-$ periodicity. In order to maintain the three-layer periodicity, it requires the alternating of $T_d\uparrow$, 1T'-I, $T_d\downarrow$ and 1T'-II, which is unlikely to occur since the nucleation of 1T'-I (1T'-II) inside the existing 1T'-II (1T'-I) at low-temperature is unfavored. On the other hand, the simultaneous nucleation of $T_d\uparrow$ and $T_d\downarrow$ domains inside single 1T'-I domain is possible as long as a change of periodicity occurs as shown in **e**. Experimentally, as shown in Fig. 2f, the periodicity does change within a single twin domain, which implies the possibility of the nucleation of opposite $T_d$ domains

a consequence, a thin planar $T_d$ unit ($-M+$) emerges owing to the crystallographic glide. The existence of a $T_d$ unit can also be understood in our notation in which the meeting region of 1T'-II ($--$, indicated by purple dashed lines between interlayers) and 1T'-I ($++$, indicated by orange dashed lines) along the $c$ axis naturally gives a layer with $-+$ interlayer shearing (the schematic model in Fig. 2c). Thus, numerous $T_d$ mono-layer interfaces exist at the 1T' monoclinic twin walls at room temperature.

Next, we turn our attention to the temperature-driven 1T' to $T_d$ first-order phase transition, which manifests itself by resistivity anomalies with an evident thermal hysteresis[32,33,43] (Supplementary Fig. 2). A phase coexistence is expected within this hysteretic temperature window but remains little explored on meso- or nano-scales. Note that a pronounced thermopower enhancement near the phase boundary was ascribed to a significant gradient of scattering processes where real-space phase inhomogeneity may play an important role[44]. To explore the real-space phase configurations, we begin with cross-section views using in-situ cryogenic-TEM. Figure 2d–e are DF-TEM images taken in the same area at 80 K ($\ll T_c$) and 300 K ($>T_c$) after a cooling/warming cycle. In the DF-TEM images using the stronger spots (green arrows in Fig. 2a), the areas associated with the major 1T'-I phase exhibit a bright contrast while regions with minor 1T'-II and newly nucleated $T_d$ phases remain dark upon cooling (from 300 K to 80 K). Interestingly, two essentially different types of periodicities consisting of alternating bright and dark stripes can be found. First, long-range stripes corresponding to two types of 1T' twin domains, exist at 300 K (Fig. 2b, e). At 80 K, additional short-range stripes appear inside individual twin domains (Fig. 2d). Instead of the $T_d$ phase growth from the existing $T_d$ units at twin walls, abundant thin-plate-like nucleation of the $T_d$ phase occurs within individual 1T' twin domains, in agreement with the appearance of additional diffraction spots (Fig. 3a).

Therefore, there appears an intimate connection between the 1T' and $T_d$ phases, and the system enters a metastable state with significant amounts of coexisting phase domains and DWs at the thermal 1T'-$T_d$ phase transition. The spatially modulating layers contain alternating $T_d$ and 1T' phases, resembling artificial thin-film superlattices. An enlarged 80 K DF-TEM image and the corresponding line profile are shown in Fig. 2f. The $T_d$/1T' phase modulation is rather periodic and the thinnest periodicity, 4-nm, consists of 6 layers of either the $T_d$ or 1T' unit. Possible atomic models are shown in Fig. 3b–e. The experimental signature of the first-order phase transition manifests itself microscopically as a nanoscale modulation of in-phase ($++/--$) and anti-phase ($+-/-+$) interlayers with quasi-periodicity. Note that both 1T'-I and 1T'-II require a mechanical glide of the layers in opposite directions when transforming into the $T_d$ phase (Fig. 1c). The persistent phase coexistence at 80 K implies an effect of mechanical constraints applying restoring forces that tend to resist the layer-wise gliding from its initial position, particularly in our capped cross-section TEM specimen; details are given in Supplementary Note 2.

### In-plane view of polar domains and domain walls.

A further identification of polar states of these thin $T_d$ layers from cross-section views is beyond the detectability limits of our low-magnification DF-TEM technique. However, our $ab$-plane DF-TEM view reveals unambiguously the existence of two types of polar domains. Figure 4a displays the in-plane DF-TEM image of two domains with bright and dark contrasts, resulting from the non-equal diffraction intensity due to the broken space-inversion of $T_d$ phase at 80 K. Note that without initial cooling, no domains and DWs is found in any specimens at room temperature (Supplementary Fig. 3a–b). The domains with two different contrasts are associated with the $\pm c$ polar axes, but the absolute polarization direction cannot be identified in the $ab$-plane TEM view. Thus, for the sake of simplicity, we assign bright-contrast domains as $T_d\uparrow$ and dark-contrast domains as $T_d\downarrow$ in this work. The step-by-step phase transition during in-situ cooling is also provided in the sequential DF-images in Supplementary Fig. 4a–e. We also confirm the coexistence of 1T' and $T_d$ phase domains during a warming cycle (Supplementary Fig. 3b–d) in which the domain contrast of the 1T' phase remains intact in different imaging conditions. (More details are shown in Supplementary Fig. 3).

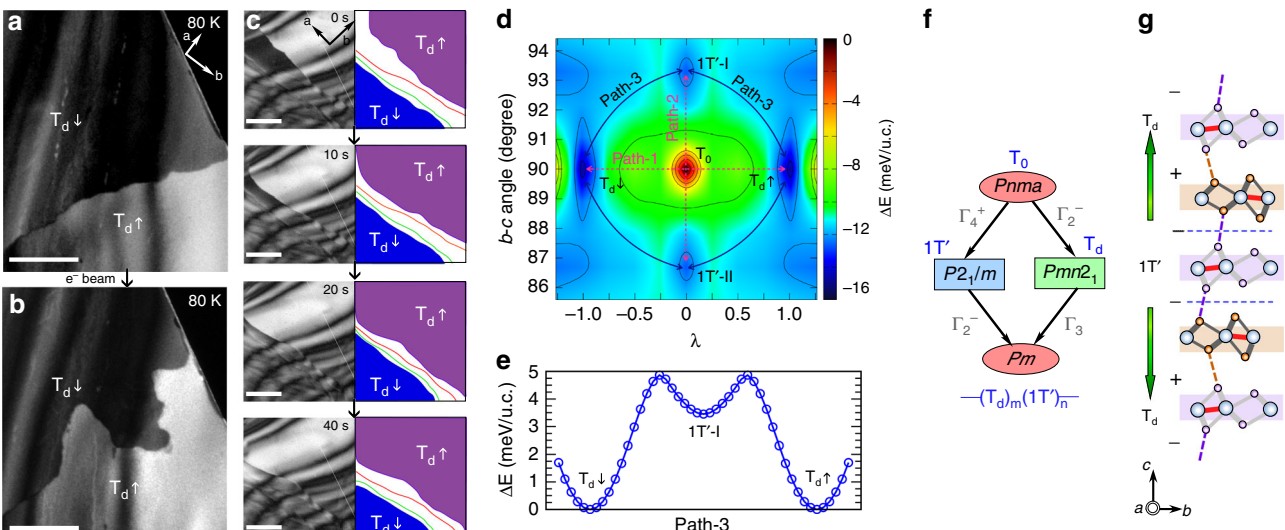

**Fig. 4** Polar domain and domain wall kinetics under e⁻ beam along plane views at 80 K. **a** DF-image of two $T_d$ domains with bright and dark contrasts. **b** The immediate DF-TEM image of the same area after exposure to a focused e⁻ beam, showing DW motion accompanying the shrinkage of the dark-contrast $T_d\downarrow$ domain. Scale bar, 500 nm. **c** Sequential snapshots obtained from the in-situ video, revealing representative polar DW motions after exposure to a focused e⁻ beam. Four visible DWs are outlined in purple, red, green and blue, which represent the DWs between gradient domains of different $T_d\uparrow/T_d\downarrow$ volume fractions along the $c$ axis. The $T_d\uparrow$ dominated (purple-shaded) domain is favored and expanded under electron beam while the $T_d\downarrow$ dominated (blue-shaded) domain has shrunk. Scale bar, 500 nm. **d** The potential energy surface as a function of the normalized interlayer displacement ($\lambda$) and **b**-**c** cell angle. The color scale denotes energy with respect to the high-energy peak $T_O$ phase. **e** The energy profile from the $T_d\downarrow$ to $T_d\uparrow$ transition along the lowest energy path-3. **f** The family tree of the crystallographic group-subgroup relation. **g** The schematic model of a $T_d\uparrow$ and $T_d\downarrow$ junction along the $c$ axis, containing one 1T′ unit as bridge. Blue dashed lines mark the phase domain walls

**Manipulation of polar domains and DWs with the electron beam**. The $T_d$ polar domains and DWs are found to be easily manipulated with in-situ e⁻ beam of the TEM at 80 K (Fig. 4a–b). The consistent and sharp domain contrast before (Fig. 4a) and after (Fig. 4b) an e⁻ beam irradiation suggests a still non-centrosymmetric structure, i.e., the $T_d$ phase. Figure 4c presents TEM snapshots from an in-situ video (Supplementary Movie 1) showing the shrinkage of a dark-contrast $T_d\downarrow$ domain, through a layer-by-layer gliding/phase-flipping process. A key feature is the observation of multi-DWs outlined by colored lines (Fig. 4c), which represent the boundaries between domains of different volume fractions of $T_d\uparrow/T_d\downarrow$ along the $c$ axis. Purple and blue shaded areas (Fig. 4c) mark $T_d\uparrow$ dominated and $T_d\downarrow$ dominated domains, respectively. The brighter contrast appears in DF-images when the $T_d\uparrow$ volume ratio is higher. The engineering of TMD polymorphs has attracted significant interest because of minimum-energy pathways or feasible transient polymorphs triggered by charge injection[45], laser irradiation[46], mechanical strain[47] and e⁻ beam irradiation[48]. In the case of MoTe₂, despite the phase change from 2H to 1T′ can occur by laser irradiation due to a local heating and Te vacancies[46,47] or electrostatic gating[45], however, any transition mechanism involving the 2H phase is excluded in our work because of the consistent electron diffraction pattern and domain contrast before and after an e⁻ beam irradiation (Fig. 4a, b and Supplementary Fig. 4f, g). Notably, the induced domains and DWs return to their original morphology after spreading a focused beam. A restorative DW motion is captured by an in-situ video (Supplementary Movie 2). The reversibility of $T_d\uparrow$ and $T_d\downarrow$ domains proves that there is no massive Te atom loss or damage by the knock-on effect during the exposure. The e⁻ beam induced domain behavior is known in ferroelectric insulators, and attributed to positive specimen charging in insulating materials[49,50]; however, no static charge accumulation is expected in semimetallic MoTe₂.

To understand the switching phenomena, we compute the potential energy landscape of MoTe₂ using first-principles density functional theory (DFT) calculations[51–55] (see Methods section). As shown in Fig. 1a, the P and M layers are related by the symmetry operation $M_z|(\frac{b}{2} + \lambda)$, where $M_z$, $b$ and $\lambda$ represent a vertical mirror, lattice vector, and the interlayer displacements between two neighboring layers, respectively. We identify a new high-symmetric orthorhombic structure $T_0$ (*Pnma*, space group #62) of MoTe₂ at $\lambda = 0$ (Fig. 1b), which belongs to the high-energy peak on the potential energy landscape of MoTe₂ (Fig. 4d). The $T_0$ phase has two instabilities: (1) an unstable in-plane optical phonon mode at the Brillouin zone center, and (2) an elastic instability yielding negative elastic stiffness coefficients (Supplementary Fig. 5). The first instability leads to an interlayer displacement of neighboring layers, yielding a double-well potential energy profile with two local minima at $\lambda = \pm0.5$ Å, representing $T_d\uparrow/T_d\downarrow$ phases. The second instability causes a rigid shear of the orthorhombic unit cell, making the $b$-$c$ cell angle non-orthogonal. By rotating the $b$-$c$ angle of the $T_0$ phase, we again obtain a double-well potential energy profile having two local minima at 86.4° and 93.6° corresponding to 1T′-I and 1T′-II phases, and the predicted monoclinic angle is in good agreement with the experimental data (93.5°−93.9°)[33,41,56]. This monoclinic distortion has the effect of shifting the neighboring layers horizontally, by about the same distance as in the $T_d$ phases, suggesting that it is driven by the same underlying microscopic instability.

Figure 4d shows the potential energy surface of MoTe₂ in the vicinity of the $T_0$ phase as a function of $\lambda$ and the $b$-$c$ angle. We obtain four minima corresponding to $T_d\uparrow$, $T_d\downarrow$, 1T′-I, and 1T′-II phases, where the $T_d$ phases are the lowest in energy with reference to the high-energy point $T_0$. A direct structural transition from the $T_d\uparrow$ to $T_d\downarrow$ (1T′-I to 1T′-II) phase through the peak along path-1 (path-2), as shown in Fig. 4d, requires overcoming a large energy barrier of height 16.9 meV/u.c. (13.5 meV/u.c.). However, there are lower-energy pathways with an energy barrier of ~5 meV/u.c., marked as path-3 in Fig. 4e, suggesting that the $T_d\uparrow$ to $T_d\downarrow$ polar converting via an

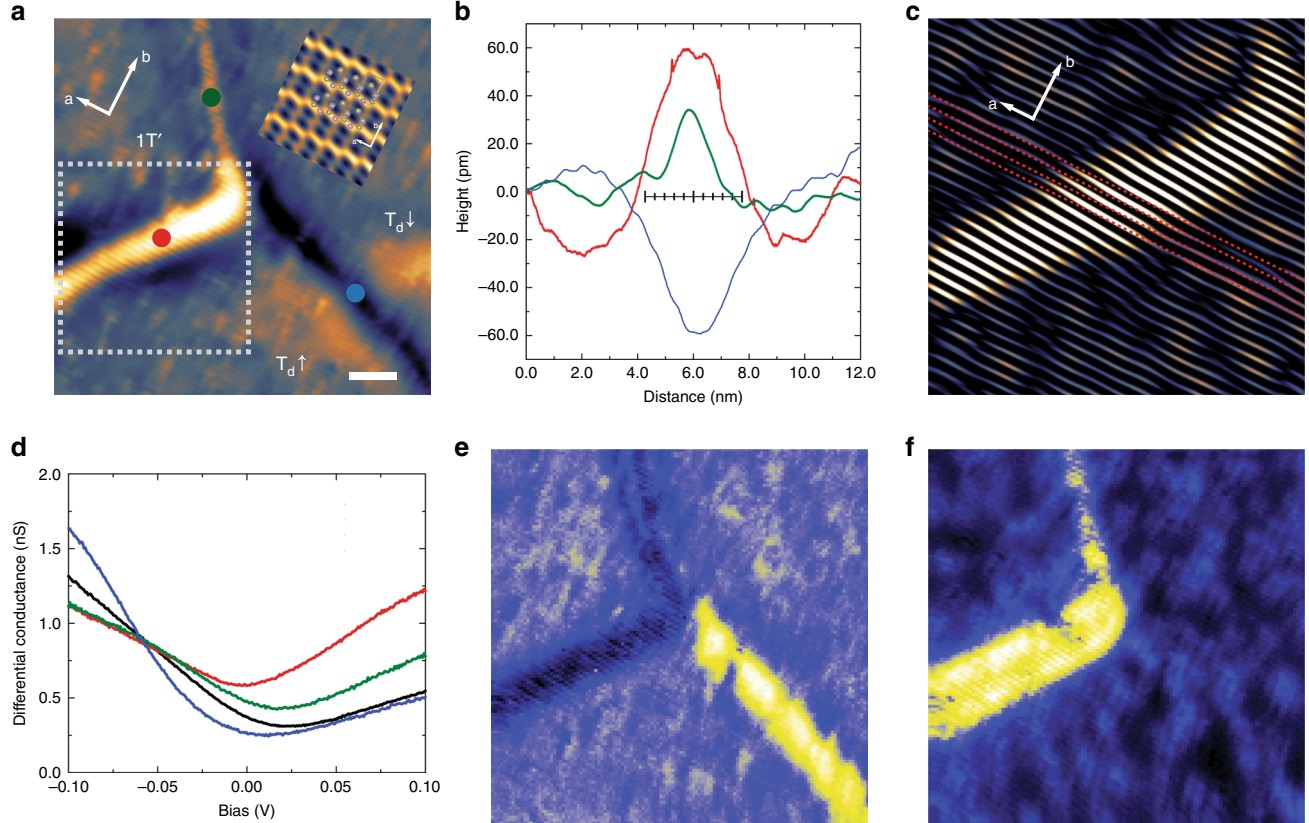

**Fig. 5** (double columns) STM spectroscopic features of lateral phase domains and DWs in MoTe$_2$:Fe at 77 K. **a** STM topography showing a junction of three domains and DWs. Inset: atomic resolution image of MoTe$_2$ showing Mo–Te–Mo chains. Scale bar: 5 nm. **b** Height profile across each of three DWs obtained from red, green and blue dots in **a**. Red and green represent the first-type protruded DWs and blue belongs to the second type. A ruler at the center shows the length of ten-unit cells for comparison. **c** Fourier filtered topography of the first type DW from the dashed rectangle in **a**. Red dashed lines mark ideal chain directions and the topography on the protruded area reveals the deviation of chain from the ideal straight line. **d** Differential conductance obtained from each DW in **a**. Red and green curves: the first-type protruded DWs; Blue curve: the second-type depressed DW; Black: the averaged curve obtained inside a domain, normalized at −100 mV, 100 pA. **e**, **f** Spatial mapping of differential conductance at −100 mV (**e**) and +100 mV (**f**), normalized at −50 mV, 100 pA

intermediate nonpolar 1T′ phase is energetically preferable as shown in Fig. 4g. In this respect, a feasible low energy path through the 1T′ DW-mediated switching process may be involved in our e⁻ beam effect. The electron beam is certainly required to trigger the layer shearing.

We next consider the T$_0$, 1T′, and T$_d$ phases from the view of symmetry. Figure 4f illustrates the MoTe$_2$ "family-tree" of the crystallographic group-subgroup relations[57]. The 1T′ and T$_d$ phases reveal that a proper transition drives from the high-symmetric T$_0$ upon the $\Gamma_4^+$ or $\Gamma_2^-$ zone center instabilities, which is consistent with the phonon dispersion shown in Supplementary Fig. 5. A detailed symmetry analysis further indicates that *Pm* (space group #6) is a subgroup of both 1T′– ($P2_1/m$) and T$_d$-MoTe$_2$ ($Pmn2_1$) and it is expected to link the 1T′ and T$_d$ phases as shown in Fig. 4f. Space group *Pm* is, indeed, the symmetry to describe those superlattice–like structures appearing across transition (Fig. 3b), providing a complete unified symmetry description of MoTe$_2$.

**Phase domain wall conductance**. Finally, the atomic-scale electronic properties of lateral polar and phase DWs are also investigated by scanning tunneling microscopy (STM). In order to increase the density of polar and phase domains and DWs in the *ab*-plane, we have grown Fe-doped MoTe$_2$ crystals (see Methods section and Supplementary Fig. 2). One polar/phase junction among T$_d$↑, T$_d$↓ and 1T′ domains is found at 77 K (Fig. 5a) in a

MoTe$_2$:Fe crystal with a slightly lower phase transition temperature (Supplementary Fig. 2). Consistent with the identical nature of each layer of 1T′ and T$_d$ phases discussed above, three domains near the junction present similar topography and spectroscopic features, as well as quasiparticle interference patterns, which are dominated by the atomic distribution of Fe dopants (the details are given in Supplementary Fig. 2 and Supplementary Fig. 6). On the other hand, DWs reveal two different types; the first type is marked with red and green dots in Fig. 5a–b), which deviates from the zigzag direction, i.e., the *a*-axis direction. The zigzags, corresponding to high-intensity lines in Fig. 5c, shift in the direction perpendicular to zigzags at the protruded area (red-dot DW) (Fig. 5c). The second type DW follows the zigzag direction (blue curve in Fig. 5a–b). The shift at the first-type DWs is likely due to the mismatch of the unit cell between the monoclinic 1T' and orthorhombic T$_d$ phase domains. These considerations lead to the most likely domain assignment shown in Fig. 5a. From our TEM and STM observations, we find that polar DWs tend to be parallel along the zigzag direction while phase DWs tend to be highly curved (Fig. 4c and Supplementary Fig. 4).

Interestingly, from tunneling spectroscopy measurements, we observe characteristic local density of states at these two types of DWs, distinct from that of the bulk (Fig. 5d). The first-type protruded DWs, namely 1T′/T$_d$ phase DWs, (red and green curves in Fig. 5d) show an enhanced conductance in the empty

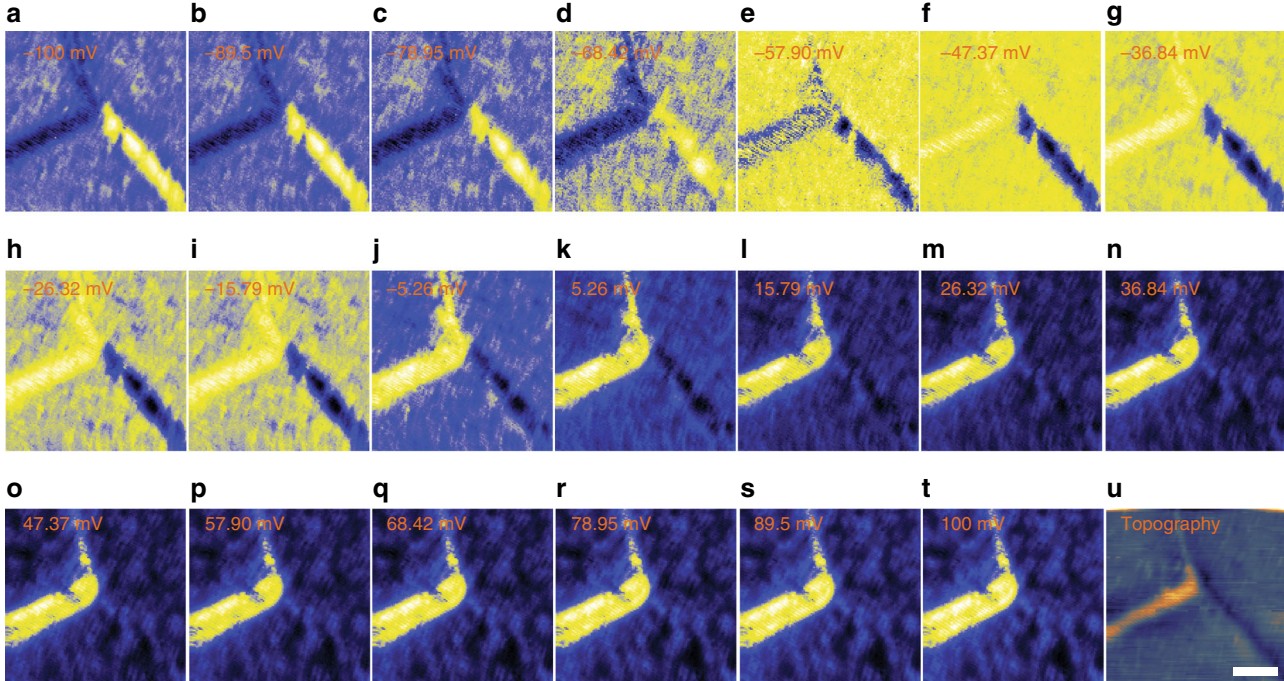

**Fig. 6** (double columns) Spatial variation of local density of state near the junction at 77 K. **a–t** A set of differential conductance maps obtained near a junction of three domains and DWs. All of spectra are normalized to the same tunneling resistance with −50 mV and 100 pA. Differential conductance spectra are obtained by demodulated lock-in signal using 10 mV of oscillation at 20 different biases from −100 to 100 mV. Twenty lock-in measurements are interlaced in between each line-scan of the topography shown in **u** to minimize the effect of thermal drift. There appear two different characteristics from the DWs. Two DWs on the upper left corner show depressed local density of state at energies below −50 mV, and start to show enhanced conductance above −50 mV. On the contrary, the DW in the right bottom corner shows the opposite feature, i.e., enhanced below −50 mV and suppressed above −50 mV. **u**, STM topography of the same area with the 40 × 40 nm$^2$ image size. Scale bar: 10 nm

state while the second-type DW (blue curve in Fig. 5d) does in the filled state. These features are also apparent in spatial mapping of conductance taken at filled (−100 mV, Fig. 5e) and empty (+100 mV, Fig. 5f) states. The systematic studies of spatial variation of local density of states are given in Fig. 6. Our results demonstrate distinct electronic properties at those polar and phase DWs in MoTe$_2$. Note that 1T′-MoTe$_2$ was earlier considered to be a topologically trivial material[16,17,41] based on the Fu-Kane $Z_2$ index criterion[58], however, recent theoretical works predict that 1T′-MoTe$_2$ inherits a higher–order topological phase featuring topologically protected 1D hinge modes at the edges[35–37]. We notice that a considerably large conductance at the first-type 1T′/T$_d$ phase DWs with the orientation-dependent feature (red and green curves in Fig. 5d). Alternatively, those protruded type 1T′/T$_d$ DWs can be promising candidates for the conducting hinge state studies in future[59].

## Discussion

In summary, for the first time we report the existence of polar domains and abundant superlattice-like arrangements of phase DWs in MoTe$_2$ using in-situ cryogenic TEM along planar and cross-section views. We also discuss the feasible low-energy pathways of the polar domain switching. Our observations open up several important directions for future exploration. First, notably, the T$_d$ polar phases of MoTe$_2$ host topologically non-trivial Weyl points[16,17,40,41]. Since T$_d$↑ and T$_d$↓ polar phases are related by the space-inversion symmetry, Weyl points in these phases will have the same location in the energy and momentum space (but opposite chirality) and are hence considered "topologically identical." One naturally expects quantum phenomena occurring due to the projection of opposite pairs of Weyl points and the resulting Fermi arc patterns at the T$_d$↑/T$_d$↓ polar DWs.

For example, as we tune this interlayer displacement parameter, $\lambda$, opposite Weyl points move towards each other in the momentum space, and finally mutually annihilate at $\lambda = 0$, i.e., no Weyl points for the T$_0$ phase. A similar manipulation of Weyl point separation and Weyl point number by interlayer displacements has been discussed in WTe$_2$[31]. In contrast to polar DWs, phase DWs are the interfaces between "topologically distinct" phases: topologically nontrivial WSM and higher-order topological phases (i.e., T$_d$/1T′ superlattice structures along the $c$ axis). Our STM observations, which imply the possible presence of conducting hinge states in the 1T′/T$_d$ phase DWs, call for further attention. Second, those T$_d$/1T′ superlattice regions with abundant phase DWs can be described as a transient state, which may be a rich area for macro-scale ordering by modulating the interlayer stacking and topological invariant. Lastly, the existence of polar domains and the electron beam manipulation of those polar DWs offer the possibility of rapid/controllable topological switching through electronic/optical excitations[31] and could be extended to other WSM or polar metals.

## Methods

**Sample preparation.** 1T′-MoTe$_2$ single crystals were grown using the flux method. Well ground Mo (Alfa Aesar, 99.9%) and Te (Alfa Aesar, 99.9%) powders were mixed with sodium chloride (NaCl, Alfa Aesar, 99.9 %) in an alumina cru-cible, which was sealed in a quartz tube under vacuum. Crystallization was con-ducted from 1100 to 960 ºC for 12 h, following a 0.5 ºC/h cooling rate to 960 ºC and then a rapid cooling to room temperature by placing the quartz tube in water (quenching). Ribbon-like crystals (3*0.5*0.1 mm$^3$) with shiny surfaces were obtained. 1T′-MoTe$_2$:Fe single crystals were grown using a similar process with the starting composition of Fe$_{0.3}$MoTe$_2$, but tends to form ten times smaller in size and less cleavability. From the analysis of STM images, the resulting estimation of the real composition of Fe impurities is ~1.06% (the details are given in Supplementary Fig. 2). The electrical transport measurements (along the $b$ axis) were taken with the standard four-probe technique using Au paste as electrodes. Temperature was

controlled by using a Physical Properties Measurement System (PPMS-9, Quantum Design), are consistent with the results in literature[32,33].

**TEM measurements.** Crystal structure, electron diffraction and domains were examined by transmission electron microscopy (TEM) in side view and plane view. Plane-view specimens were obtained by scotch-tape exfoliation, while side-view specimens were fabricated as follows. First, two silicon slabs and one $MoTe_2$ thin plate were clamped and glued together using epoxy bond (Allied, Inc) with sides facing each other to make a sandwich structure. The $MoTe_2$ sandwich was further thinned down by mechanical polishing, followed by Ar-ion milling, and studied using a JEOL-2010F field-emission TEM quipped with a low-$T$ sample stage and a room-$T$ double-tilt sample stage. We observed in-plane polar domains by DF-TEM imaging taking $g_1 \pm = \pm (1, 2, \bar{1})$ spots along the [101] direction, ~14° tilting from the [001] zone and the side-view twin domains using $g_2 \pm = \pm (1, \bar{1}, 2)$ spots along the [110] direction, ~60° tilting from the [100] zone. HAADF-STEM imaging with atomic-column resolution was carried out using the field-emission JEOL-2100F microscope equipped with a spherical aberration Cs corrector. All images are raw data. HAADF-STEM images were acquired in two conditions: $512 \times 512$ with 0.019 nm and 0.015 nm/pixel with collection angle between 80–210 mrad.

**STM measurements.** STM and spectroscopy measurements were performed at liquid nitrogen temperature using a Unisoku ultra-high vacuum SPM System (USM-1500) with a cleaving stage in the chamber. A Cu(111) sample that is cleaned by repeated cycles of sputtering and annealing prior to scanning has been used as a reference sample. A Pt/Ir tip is heated by electron beam bombardment in ultra-high vacuum condition to remove contaminations from air, and further treated on Cu(111) sample until it shows a metallic conductivity and the Cu(111) surface state spectroscopy. Fe doped $MoTe_2$ sample is fixed at a sample plate by silver epoxy (Epotek H20E) and a metal post is attached to the top with the same epoxy. Then the sample is introduced to ultra-high vacuum chamber and cleaved at room temperature in the cleaving stage followed by insertion to $LN_2$ cooled STM head. Differential conductance is measured by modulation of bias and demodulation of tunneling current using lock-in technique ($f = 611$ Hz, 10 mV with AC added to the bias).

**Theoretical calculations.** All the first-principles DFT calculations were performed using the Vienna ab initio simulation package (VASP) within the projected-augmented wave (PAW) framework[51,52]. We considered 6 valence electrons of Mo ($4d^5 5s^1$) and 6 valence electrons of Te ($5s^2 5p^4$) in the PAW pseudopotential. We used the PBEsol exchange-correlation functional to treat exchange and correlation effects[53]. A Monkhorst-Pack $k$ mesh of size $8 \times 12 \times 4$ was used to sample the $k$-space, and 600 eV was used as the kinetic energy cutoff of the plane wave basis set. We also considered effects due to the on-site Coulomb interaction of Mo $4d$ electrons, which were recently reported to be crucial in the precise determination of electronic structure of $MoTe_2$ near the Fermi-level. Within the DFT + U scheme, we used U = 2.8 eV and J = 0.4 eV to simulate Mo $4d$ electrons at the mean-field level. These values are reported to correctly describe the topological phase transitions and bulk electronic band structure of $MoTe_2$ near the Fermi-level[54]. The structures were optimized until the Hellmann-Feynman residual forces were less than $10^{-4}$ eV/Å, and $10^{-9}$ eV was defined as the convergence criterion for the electronic self-consistent calculations. Optimized lattice parameters and structural details are given in the Supplementary Table 1. Given the symmetry of 1T′ and $T_d$ structures, $a$ and $b$ lattice vectors are interchangeable. In the DFT calculations, we used a convention in which Mo–Mo zigzag run along the $b$-lattice vector ($a > b$). The phonon calculations were performed using the finite-difference approach as implemented in the VASP software. Supercell of size $2 \times 4 \times 1$ was used for phonon calculations, and PHONOPY code was used for the post-processing of phonons[55]. All the inner-coordinates of atoms were fully optimized, except for the modulated structures along the unstable phonon mode at Γ, which are optimized while keeping the coordinates of atoms frozen in the direction of modulation vectors. However, $T_d\uparrow$ and $T_d\downarrow$ structures were further optimized without any constraints.

## Data availability

The authors declare that all source data supporting the findings of this study are available within the article and the Supplementary information file.

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

## Acknowledgements

The work at Rutgers was funded by the Gordon and Betty Moore Foundation's EPiQS Initiative through Grant GBMF4413 to the Rutgers Center for Emergent Materials and by NSF DMREF Grant No. DMR-1629059.

## Author contributions

F.T.H. conducted the TEM experiments. S.J.L. carried out the STM observations. S.S., J.K., K.R. and D.V. carried out the theoretical analysis. J.W.K. performed transport measurements. L.Z. synthesized single crystals. F.T.H. and M.W.C performed the STEM observations. F.T.H., S.J.L., S.S., D.V., and S.W.C. wrote the paper. S.W.C. initiated and supervised the research.

## Additional information

**Competing interests:** The authors declare no competing interests.

