## [Peer Review File · Nature Communications]

Reviewers' comments:

Reviewer #1 (Remarks to the Author):

The authors report on a detailed electron microscopy and first-principles calculations study on the domain-wall structures in MoTe₂. Given the recent interest in metals without inversion symmetry, particularly concerning Weyl physics and “ferroelectric metallicity,” I find that the paper does not provide significantly new contributions to these fields to warrant publication in Nature Communications. First, the major claim is the discovery of “intriguing” DWs in the low-T phase of Weyl MoTe₂, which is not too surprising given the fact that they are required to appear because of the change in Laue class across the transition. Although MoTe₂ is a Weyl semimetal, which makes it of broad topical interest, it belongs to the general class of polar metals it certainly belongs to the polar metals as pointed out on line 58 by the authors. Secondly, and to this point, this study is not so different from two previous studies on DWs in the polar metal Ca₃Ru₂O₇, appearing in Nano Lett., 2018, 18 (5), pp 3088–3095 and PRB 99, 014105 (2019). Interestingly although these studies are the first to report on DWs in polar metals, they are omitted from the reference list in the present manuscript. The DWs that Huang et al. report exhibit many similarities with that shown in PRB 99, 014105 (2019). In particular, the junction in fig. 4 is similar to the t-junction already reported for the polar metal Ca₃Ru₂O₇. In addition, experiments to move the domain wall are also reported in the Nano Lett. paper. At a minimum these papers need to be referenced and discussed in the context of the binary MoTe₂. This manuscript does propose a model for switching based on DFT simulations, which I find interesting, but that alone is not sufficient to warrant publication. I was hoping to learn more about implications on Weyl physics due to the DW geometry --- as the authors point out “Our results open new avenues for investigating topological interfacial states” but they do not elaborate on this aspect other than suggesting the presence of hinge states. If so, then the work could be suitable for publication since it would go beyond what is presently reported in the literature. For the above reasons, while the work is carefully performed and described, I do not think this study is sufficiently novel to warrant publication in Nat. Comm.

Reviewer #2 (Remarks to the Author):

I liked this paper quite a bit and I think it should be published. Just a few comments:

1. there were a few english related typos that should be fixed.
2. The authors seemed to be treating the Fe substitution quite trivially. Is this really the case? where does the Fe go and how does it behave electronically? Some explanation for this should be given.

After this, the paper seems to be in good shape.

Reviewer #3 (Remarks to the Author):

This is a high quality work, one of the few that describes the atomic scale structure of domain walls in topological polar semimetals.

Personally, I found the discovery that the 1T' walls have T_d character interesting. The transformation of T_d domains from up to down occurs through an intermediate 1T' phase. The corresponding switching pathway calculation confirming this is interesting.

The discussion of the structure is hard to follow. In Figure 1, I might suggest adding “x” for inversion

centers, so we can see where they are present and where they are broken.

Is there a value for polarization in the polar Td phase from experiments and from theory?

Reply to the comments of reviewers. The new aspects and novelty of our results are clearly stated in our revised manuscript. All changes in the manuscript and supplementary information are highlighted in blue.

Reviewer #1

(1) *“The authors report on a detailed electron microscopy and first-principles calculations study on the domain-wall structures in MoTe₂. Given the recent interest in metals without inversion symmetry, particularly concerning Weyl physics and “ferroelectric metallicity,” I find that the paper does not provide significantly new contributions to these fields to warrant publication in Nature Communications. First, the major claim is the discovery of “intriguing” DWs in the low-T phase of Weyl MoTe₂, which is not too surprising given the fact that they are required to appear because of the change in Laue class across the transition. Although MoTe₂ is a Weyl semimetal, which makes it of broad topical interest, it belongs to the general class of polar metals it certainly belongs to the polar metals as pointed out on line 58 by the authors. Secondly, and to this point, this study is not so different from two previous studies on DWs in the polar metal Ca₃Ru₂O₇, appearing in Nano Lett., 2018, 18 (5), pp 3088–3095 and PRB 99, 014105 (2019). Interestingly although these studies are the first to report on DWs in polar metals, they are omitted from the reference list in the present manuscript. The DWs that Huang et al. report exhibit many similarities with that shown in PRB 99, 014105 (2019). In particular, the junction in fig. 4 is similar to the t-junction already reported for the polar metal Ca₃Ru₂O₇. In addition, experiments to move the domain wall are also reported in the Nano Lett. paper.”*

Reply: We appreciate reviewer's valuable time and efforts in evaluating our manuscript. As pointed out by the reviewer, MoTe₂ being a polar Weyl semimetal, concerning Weyl physics and ferroelectric metallicity will definitely draw a significant attention in a broad range of readers. In particular, the recent discoveries of ferroelectricity in monolayer MoTe₂ (Yuan *et al. Nature communications* 2019) as well as in few-layer WTe₂ (Fei *et al. Nature* 2018), and the terahertz light induced ultrafast symmetry switch in WTe₂ (Sie *et al. Nature* 2019), which has been proposed to tune the topological invariants, make our current work timely. However, he/she raised questions on the novelty of our work in a not-so-clear manner.

First, we fully agree that domains and DWs are required to appear by symmetry breaking —space inversion symmetry breaking in MoTe₂. As stated in the introduction, the symmetry argument was one of the motivating factors of our work. Since Weyl semimetals must have broken space inversion or time reversal symmetry, they must have domains and domain walls. However, numerous papers on Weyl semimetals in highly prestigious journal have been published without revealing the structure of structural/magnetic domains and domain walls in those materials [Xu *et al. Science* (2015); Deng *et al. Nat. Phys.* (2016); Huang *et al. Nat. Mater.* (2016); Shekhar *et al. Nat. Phys.* (2015); Jiang *et al. Nat. Commun.* (2017); Wu *et al. Nat. Phys.* (2017) and more], even though domains and domain walls presumably have direct consequences on the nature of topologically protected surface states.

In this work, we provide the first experimentally demonstration of phase domains/DWs as well as polar domains/DWs in polar Weyl semimetal MoTe₂. The demonstration is not just finding domains/DWs, but also brings new surprisingly insights into unveiling the intriguing Weyl physics. For example, we found the presence of conducting hinge states in those phase DWs where phase DWs are the interfaces between “topologically distinct” phases: topologically nontrivial WSM and high-order topological phases (i.e. T_d/1T' superlattice structures along the *c* axis). The presence of the topological hinge states makes the reported phase DWs unique and fundamentally important. In contrast to phase DWs, polar DWs are the interfaces between “topologically identical” phase. Since T_d↑ and T_d↓ polar phases are related by the space-inversion symmetry, Weyl points in these phases will have the same location in the energy and momentum space, but opposite chirality and are hence considered “topologically identical.” One naturally expects quantum phenomena occurring due to the projection of opposite pairs of Weyl points and the resulting Fermi arc patterns at the T_d↑/T_d↓ polar DWs. For example, as we tune this interlayer displacements parameter, λ , opposite Weyl points move towards each other in the momentum space, and finally mutually annihilate at $\lambda = 0$, i.e. no Weyl points for the T₀ phase. A similar manipulation of Weyl point separation and Weyl point number by interlayer displacements has been discussed in WTe₂³¹. In addition, these polar domains and DWs can be still controllable. We emphasize that those discoveries provide new research opportunities not only on a large number of *polar topological semimetals* (e.g. hexagonal ABC compound such as LiZnBi, doped HgCr₂(Se,Te)₄ and Ta₃S₂, Cd₂As₃, TaIrTe₄, (W, Mo)(Te,P)₂, family, (Ta, Nb)(As, P) and RAlGe family) but also on *chiral and/or*

magnetic topological materials owing to the ubiquitous symmetry breaking nature. The broad impacts of our work in searching for “new topological interfacial states” of domains/DWs are highly anticipated.

On the other hand, he/she made the following unclear comparisons with $\text{Ca}_3\text{Ru}_2\text{O}_7$, “*this study is not so different from two previous studies on DWs in the polar metal $\text{Ca}_3\text{Ru}_2\text{O}_7$.*” We, first, thank the reviewer for bringing out $\text{Ca}_3\text{Ru}_2\text{O}_7$ and two mentioned papers which are closely related to one of our core topics—polar semi(metal). Beyond just citing these papers, we highlight the achievements of these works in the introduction of our revised manuscript. However, we emphasize that the origins of ferroelectricity between $\text{Ca}_3\text{Ru}_2\text{O}_7$ and MoTe_2 are fundamentally distinct. $\text{Ca}_3\text{Ru}_2\text{O}_7$ belongs to a hybrid “improper” ferroelectric (HIF) family with polarity as a secondary effect, whereas $T_d\text{-MoTe}_2$ can be referred to a “proper” ferroelectric with zone-center symmetry breaking instabilities leading to the ferroelectric transition (please see Supplementary Fig. S7 for the phonon dispersion). Clearly, the resultant domain configurations vary significantly. For example, the T-junction mentioned by the reviewer is a junction among polar domains with different *ab*-plane directional variants while our STM results shown in Fig. 4 reveal the lateral phase domains/DWs between $1T'$ and $T_d\uparrow/T_d\downarrow$. Again, the presence of the topological hinge states makes the reported phase DWs fundamentally important. As far as we notice, there is no report of any structural phase transition and/or no hint of the phase domains in $\text{Ca}_3\text{Ru}_2\text{O}_7$. Consistently, the unpublished results of our own extensive study of $\text{Ca}_3\text{Ru}_2\text{O}_7$ domains/DWs do not show any of those.

Next, we do not understand the meaning of these statements: “*In particular, the junction in fig. 4 is similar to the t-junction already reported for the polar metal $\text{Ca}_3\text{Ru}_2\text{O}_7$. In addition, experiments to move the domain wall are also reported in the Nano Lett. paper.*” Polar metals have drawn much attention in the ferroelectric community partially because of the switching dilemma. It is fascinating but no surprising that the external strain can be an alternative way to manipulate polar interlocked ferroelastic domains in orthorhombic $\text{Ca}_3\text{Ru}_2\text{O}_7$. On the other hand, the switching mechanism in orthorhombic $T_d\text{-MoTe}_2$ semimetal by electron beam in bulk form (TEM specimen is ~ 35 nm thick) is totally unexpected. This is the very first report showing clear real-space ferroelectric reversible switching process in a Weyl semimetal (please check Supplementary Movie 1 and 2). It turns out that a feasible low energy path through the $1T'$ DW-mediated switching process may be involved as supported by

first-principle calculations. The surprising elegant structural and energetic pathway connection between the $1T'/T_d$ phases and domain walls is further explained in the reply 2 below. In summary, we emphasize that the underlying physics between $\text{Ca}_3\text{Ru}_2\text{O}_7$ and MoTe_2 certainly differs and deserves dedicated discussions.

(2) "At a minimum these papers need to be referenced and discussed in the context of the binary MoTe_2 . This manuscript does propose a model for switching based on DFT simulations, which I find interesting, but that alone is not sufficient to warrant publication. I was hoping to learn more about implications on Weyl physics due to the DW geometry --- as the authors point out "Our results open new avenues for investigating topological interfacial states" but they do not elaborate on this aspect other than suggesting the presence of hinge states. If so, then the work could be suitable for publication since it would go beyond what is presently reported in the literature. For the above reasons, while the work is carefully performed and described, I do not think this study is sufficiently novel to warrant publication in Nat. Comm."

Reply: We sincerely thank the reviewer's encouraging acknowledgement of our work: "*the work is carefully performed and described*" and "*...suggesting the presence of hinge states. ...since it would go beyond what is presently reported in the literature*". We would also like to note that one of the other reviewers (#3) also recognized the novelty of our work by stating: "*This is a high quality work, one of the few that describes the atomic scale structure of domain walls in topological polar semimetals. Personally, I found the discovery that the $1T'$ walls have T_d character interesting. The transformation of T_d domains from up to down occurs through an intermediate $1T'$ phase. The corresponding switching pathway calculation confirming this is interesting.*" As we explained in the above reply, our work paves the way for gaining an even deeper knowledge and unveiling multifunctional aspect of polar Weyl semimetals MoTe_2 such as the existence of polar DWs, phase DWs, the possibility of topological switching through electronic/optical excitations, and the presence of hinge states of phase DWs. We would like to further elaborate our work from the group-subgroup analysis in the following.

The discovery of the transient state in the form of superlattice-like structures with abundant atomically sharp phase DWs along the c axis is a big surprise. To resolve the superlattice-like-structure puzzle, we have employed the group-subgroup analysis using the isotropy subgroups representing the space groups allowed by spontaneous symmetry breaking phenomena within the framework of Landau theory of phase transitions [H. T. Stokes and D.

M. Hatch, Isotropy subgroups of the 230 crystallographic space groups (World Scientific Publishing, 1988), ref. 57 in the main text]. A detailed group-theoretical analysis reveals that Pm (space group #6) is a subgroup (and the only common subgroup) of both $1T'$ ($P2_1/m$, space group #11) and T_d - MoTe_2 ($Pmn2_1$, space group #31) and it is expected to link the $1T'$ and T_d phases (see below for the symmetry tree). Space group Pm is, indeed, the symmetry to describe those superlattice-like structures as shown below (as well as shown in Supplementary Fig. S4), providing a complete unified symmetry description of MoTe_2 .

A proper transition drives from the high-symmetric orthorhombic structure T_0 ($Pnma$, space group #62) to either nonpolar $1T'$ or polar T_d phase upon the Γ_4^+ or Γ_2^- zone center instabilities, respectively. The group-subgroup analysis presented below fully agrees with the low DW-energy switching path shown in the potential energy landscape. Notably, such elegant crystallographic connection between two competing phases *via* an intermediate symmetry state is very rare in the literature. It has only been found in the superconducting $\text{La}_{2-x}\text{B}_x\text{CuO}_4$ system ($B = \text{Ba}, \text{Sr}$) (*Physica C*, **206**, 183, 1993), at the so-called morphotropic phase boundaries (MPB) of ferroelectric $\text{PbZr}_{1-x}\text{Ti}_x\text{O}_3$ (*PRB*, **63**, 014103, 2000) and topological defects in h - $\text{In}(\text{Mn},\text{Ga})\text{O}_3$ manganites (*PRL* **113**, 267602, 2014). The intimate relation of $1T'$ and T_d phases might explain the filamentary superconductivity that many groups have observed in $1T'$ - and T_d - MoTe_2 and put them into the context of polar order in this system.

Furthermore, as pointed out by the reviewer, the presence of the hinge states of phase DWs is novel. Considering the current interest regarding the possible higher-order topology in 2D (Mo,W)Te₂ and in other 3D materials that host 1D conducting modes, such as Bi, this finding also constitutes an important added value that deserves publication in *Nature Communications*. The 1T' phase was recently predicted to be a higher order topological semimetal [refs. 35-37 in the main text] with conducting 1D hinge states. We have incorporated spatial variation of local density of state near the phase junction in the supplementary Fig. S9, where we do observe signatures of the large 1D conductance at the boundaries of the 1T' phase, which is missing at the boundaries of the T_d -domains. This is an encouraging experimental signature revealing the higher-order topology present in 1T'-MoTe₂. Please note that so far only bismuth has been experimental reported to be a higher topological insulator (technically, a semimetal). Although, we see the signatures of higher topology in 1T'-MoTe₂, this requires some more conclusive future experiments. We decided not to dig this topic in much detail in our current work, since

that would amount to a full set of interface/superlattice calculation as well as extensive STM experiments on the orientation dependent phase DWs, which is evidently beyond the scope of this work. We believe that adding those results might blur the main message of our present work, which is more focused on the structural phase transitions and the physics of polar/phase DWs in a Weyl semimetal.

Finally, in this work we not only address the subtlety of the structural phase transitions in Weyl semimetal MoTe₂ at atomic level, we also offer theoretical explanation of the observed phase transitions and phase domain wall formation, which has yet been unexplored in the polar Weyl semimetals. At the time of submission, our work was the very first report on directly probing microstructures and atom-resolved domain/DW structures in polar Weyl semimetal crystals, actually in all topological materials. During the rebuttal process, we noticed that ferroelectric domains have been resolved in WTe₂ single crystal very recently (Sharma *et al.* Sci. Adv. 2019;5: eaax5080) and several other papers reporting ferroelectricity in few-layer (W,Mo)Te₂ published in *Nature* and its subsidiary journals, as mentioned above. This clearly indicates that the study of domains/DWs in topological materials is certainly a quickly developing field and justifies the novelty and timeliness of our work.

In order to comply with the reviewer's suggestions about polar metal, Weyl physics and DW geometry as well as our new group-subgroup analysis, we modified the introduction, main text and conclusion as follows:

In page 2-3,

“Some progress has been made in, for example, the polar interlocked ferroelastic domains observed in polar metal Ca₃Ru₂O₇^{25,26} and the structural defect-mediated polar domains in metallic GeTe.²⁷ In this context, exploring the domain structures in polar Weyl semimetal would be particularly important because the Weyl points and Fermi-arc connectivity can be manipulated via domain reorientation or locally modified order parameters at these DWs²⁸⁻³¹. Solving the Weyl equation under the experimentally known DW geometry is highly desired.”

In page 8,

“We next consider the T_0 , $1T'$ and T_d phases from the view of symmetry. Figure 3d illustrates the MoTe_2 “family-tree” of the crystallographic group-subgroup relations⁵⁷. The $1T'$ and T_d phases reveal that a proper transition drives from the high-symmetric T_0 upon the Γ_4^+ or Γ_2^- zone center instabilities, which is consistent with the phonon dispersion shown in Supplementary Fig. S7. A detailed symmetry analysis further indicates that Pm (space group #6) is a subgroup of both $1T'$ ($P2_1/m$) and $T_d\text{-MoTe}_2$ ($Pmn2_1$) and it is expected to link the $1T'$ and T_d phases as shown in Fig. 3d. Space group Pm is, indeed, the symmetry to describe those superlattice-like structures appearing across transition (Supplementary Fig. S4), providing a complete unified symmetry description of MoTe_2 .”

In page 9-10,

“First, notably, the T_d polar phases of MoTe_2 host topologically non-trivial Weyl points^{16,17,40,41}. Since $T_d\uparrow$ and $T_d\downarrow$ polar phases are related by the space-inversion symmetry, Weyl points in these phases will have the same location in the energy and momentum space (but opposite chirality), and are hence considered “topologically identical.” One naturally expects quantum phenomena occurring due to the projection of opposite pairs of Weyl points and the resulting Fermi arc patterns at the $T_d\uparrow/T_d\downarrow$ polar DWs. For example, as we tune this interlayer displacement parameter, λ , opposite Weyl points move towards each other in the momentum space, and finally mutually annihilate at $\lambda = 0$, i.e. no Weyl points for the T_0 phase. A similar manipulation of Weyl point separation and Weyl point number by interlayer displacements has been discussed in WTe_2 ³¹. In contrast to polar DWs, phase DWs are the interfaces between “topologically distinct” phases: topologically nontrivial WSM and high-order topological phases (i.e. $T_d/1T'$ superlattice structures along the c axis). Our STM observations, which imply the possible presence of conducting hinge states in the $1T'/T_d$ phase DWs, call for further attention. Second, those $T_d/1T'$ superlattice regions with abundant phase DWs can be described as a transient state, which may be a rich area for macro-scale ordering by modulating the interlayer stacking and topological invariant. Lastly, the existence of polar domains and the electron beam manipulation of those polar DWs offer the possibility of rapid/controllable topological switching through electronic/optical excitations³¹ and could be extended to other WSM or polar metals.”

Revised Figure 3

Figure 3 (double columns) Polar domain and domain wall kinetics under e^- beam along plane views at 80 K. **a**, DF-image of two T_d domains with bright and dark contrasts. **b**, An immediate DF-TEM image of the same area after exposure to a focused e^- beam, showing DW motion accompanying the shrinkage of the dark-contrast $T_d\downarrow$ domain. **c**, Sequential snapshots obtained from the in-situ video, revealing representative polar DW motions after exposure to a focused e^- beam. Four visible DWs are outlined in purple, red, green and blue, which represent the DWs between gradient domains of different $T_d\uparrow/T_d\downarrow$ volume fractions along the c axis. The $T_d\uparrow$ dominated (purple-shaded) domain is favored and expanded under electron beam while the $T_d\downarrow$ dominated (blue-shaded) domain has shrunk. Scale bar, 500 nm. **d**, Top panel shows the potential energy surface as a function of the normalized interlayer displacement (λ) and $b-c$ cell angle. The color scale denotes energy with respect to the high-energy peak T_0 phase. The bottom panel depicts the energy profile from the $T_d\downarrow$ to $T_d\uparrow$ transition along the lowest energy path-3. The right panel illustrates the family tree of the crystallographic group-subgroup relation. **e**, The schematic model of a $T_d\uparrow$ (purple-shaded) and $T_d\downarrow$ (blue-shaded) junction along the c axis, containing one $1T'$ unit (yellow-shaded) as bridge.

Reviewer #2

(1) I liked this paper quite a bit and I think it should be published. Just a few comments:

Reply: We appreciate the reviewer's comments on our manuscript. Thank you!

(2) There were a few English related typos that should be fixed.

Reply: We sincerely thank the reviewer for this suggestion. The manuscript was significantly revised to reflect the reviewer's requests and read by a native English speaker.

(3) The authors seemed to be treating the Fe substitution quite trivially. Is this really the case? where does the Fe go and how does it behave electronically? Some explanation for this should be given. After this, the paper seems to be in good shape.

Reply: Regarding the Fe effect, we used a nominal ratio of 30% Fe sources in the starting material prior to the flux growth. However, we estimated the real composition of Fe impurities in grown crystals to be only ~1.06% (359 Fe atoms in a 50×50 nm² shown below). Fe doping effect is observable in transport properties. As shown below, the polar phase transition temperature decreases slightly with Fe doping. With locally-probing STM, we had difficult to find polar or phase domain/DWs in the pristine MoTe₂ while those can be found readily in 1T'-MoTe₂:Fe specimen. The topographic identification of Fe doping and quantification are shown in Methods and Supplementary Fig. S2.

Supplementary Figure 2

Supplementary Figure S2 Chemically tunable polar structural transition. **a**, Temperature dependence of the ab -plane electrical resistivity of 1T'-MoTe₂ and 1T'-MoTe₂:Fe single crystals at ambient pressure. Measurements were carried out with the electric current fixed along the b crystallographic axis. Insets show photographic images of crystals. Scale bar, 1 mm. The 1T' to T_d polar transition can be identified from the sudden decrease of resistance during cooling (blue and green curves) and the abrupt increase of resistance during warming (red and orange curves). Both resistivity curves show metallic behavior with a thermal hysteresis setting in below room T. **b**, Temperature derivative of resistivity, $d\rho(T)/dT$. Anomalous hysteresis loops are observed at 260-230 K in 1T'-MoTe₂ and 180-135 K in 1T'-MoTe₂:Fe single crystals. The phase transition can be tuned to lower temperature with Fe doping. **c**, Topographies of three typical defects found in 1T'-MoTe₂:Fe samples with scan parameters as -0.3V, 100 pA: top, a depression in the middle of a zigzag; middle, a protrusion connecting two zigzags; bottom, a protrusion in the middle of a zigzag. The first defect with the

depression is identified as a Te vacancy since we constantly observed in pristine MoTe₂ samples. The bottom two defects with protrusion features are common in 1T'-MoTe₂:Fe specimen and were assumed to be single Fe impurities present in the two distinctive Mo sites within the first layer as marked in (e). Scale bar: 1 nm. **d,e**, Estimation of Fe concentration is performed on a STM topography of 50 nm lateral size shown in (d) with scan parameters as -0.3V, 200 pA. The resulting estimation of the real composition of Fe impurities is ~1.06% (359 Fe atoms in a 50×50 nm² square area).

Reviewer #3

(1) This is a high quality work, one of the few that describes the atomic scale structure of domain walls in topological polar semimetals. Personally, I found the discovery that the 1T' walls have Td character interesting. The transformation of Td domains from up to down occurs through an intermediate 1T' phase. The corresponding switching pathway calculation confirming this is interesting.

Reply: We appreciate the reviewer's nice summary on our manuscript. Thank you!

(2) The discussion of the structure is hard to follow. In Figure 1, I might suggest adding "x" for inversion centers, so we can see where they are present and where they are broken.

Reply: We have now marked the inversion centers in the revised Figure 1. The readability has also been improved in the resubmitted manuscript. Thanks for pointing it out.

Revised Figure 1

(3) Is there a value for polarization in the polar T_d phase from experiments and from theory?

Reply: Since MoTe_2 is semimetal, the polarization is not well defined from computational perspective. We try to explain the polar distortion from the crystallographic point of view as presented below. The estimation of the net dipole moment based on the bulk T_d structure [ref. 34 in the main text] is $3.6 \times 10^{11} \text{ e}^-/\text{cm}^2$ ($=0.058 \mu\text{C}/\text{cm}^2$). Please note that recent report on monolayer MoTe_2 with the $d1T$ trimerized structure can achieve $0.68 \mu\text{C}/\text{cm}^2$ from their DFT calculation (Yuan *et al. Nature communications*, **10**, 1775, 2019). The following discussion is provided in the Supplementary section I and Supplementary Figure S1.

Supplementary Fig. S1 Schematics view of the macroscopic polarization in T_d phase. **a**, $1T'$ -I and **b**, $T_d\uparrow$ phase of MoTe_2 . Mo, blue; Te of P-layer, orange; Te of M-layer, purple. The insets show the bonding environments of the P layers. A net out-of-plane dipole moment can be induced in the T_d phase as a result of the slightly different Te environments on the top and

bottom layers. Horizontal dotted lines shown in the insets indicate the average positions of Te octahedra. Green arrows correspond to the displacements of Mo ions along the c axis. The symbol x marks the inversion center.

Supplementary section I

The origin of the uncompensated dipole in T_d - MoTe_2

In this section, we explain the polar distortion in semimetallic MoTe_2 from the crystallographic point of view. The geometric structure of centrosymmetric 1T'-I MoTe_2 with ++ interlayer pattern is illustrated in Supplementary Fig. S1a. Each Mo ion (blue spheres) sits in a distorted Te octahedron. Because of the Mo-Mo metallic bonding, Mo ions shift off the center of the distorted Te octahedra. Green arrows indicate the $\pm z$ displacement of Mo ions away from the average z positions of the neighboring Te ions. Since the top and bottom Te layers (yellow and red spheres) are confined by space inversion symmetry (the inversion center located at the Mo-Mo bond, marked at x), the $\pm c$ dipole moment is cancelled out. The intrinsic difference between 1T' and T_d arises from the interlayer gliding of the structure. As long as the interlayer gliding is arranged in either +- or -+ pattern, it will break the inversion symmetry while any additional vertical displacement is not required. The non-centrosymmetric structure is also clear when tracking the Te-Te interlayer bonding outlined by orange and purple dashed lines (Supplementary Fig. S1). The absence of any inversion center induced by the interlayer gliding correlated with the fact that $\pm c$ directions are geometrically non-equivalent. It does not necessarily accompany the existence of a net dipole moment in the sense that assuming each P or M layer is as rigid and symmetric as that of 1T' (Fig. S1a inset). Thus, any interlayer gliding is not sufficient to explain the polar origin in the T_d phase.

The inset of Fig. S1b demonstrates that the polar distortion is, in fact, caused by the local asymmetric bonding environment between the top Te (green and dark blue spheres) and bottom Te (yellow and red spheres) ions around the Mo ions. Because of the broken inversion symmetry, now the top and bottom Te are symmetry independent as indicated by different colors. Consequently, it gives additional vertical degree of freedom of those Te sites and leads to a net dipole moment along the c axis. The estimation of the net dipole moment based on the T_d structure [ref. 34 in the main text] is $3.6 \times 10^{11} \text{ e-}/\text{cm}^2$ ($=0.058 \mu\text{C}/\text{cm}^2$). Note that a recent report on monolayer MoTe_2 with the $d1T$ trimerized structure can achieve $0.68 \mu\text{C}/\text{cm}^2$ from

the DFT calculations [ref. 38 in the main text]. As discussed above, the asymmetric Te bonding environment of the T_d phase is the reason that moves the average negative center away from the Mo-Mo center even though the magnitude is almost negligible compared with traditional ferroelectric perovskite (three orders smaller than $BaTiO_3$). Despite the fact that $MoTe_2$ is semimetal, the charge distribution is expected to be highly anisotropic in the van der Waals layered structure. Therefore, the dipole-dipole interaction in the polarization direction may not be screened as a similar mechanism proposed in ferroelectric metal $LiOsO_3$ [ref. 23 in the main text]. Finally, we note that two distinct surfaces of $MoTe_2$ at low temperatures have been reported in literatures [refs. 29-30 in the main text], and they correspond to the $T_d \uparrow$ and $T_d \downarrow$ in this work.